# In Vitro and In Vivo Effect of pH-Sensitive PLGA-TPGS-Based Hybrid Nanoparticles Loaded with Doxorubicin for Breast Cancer Therapy

**DOI:** 10.3390/pharmaceutics14112394

**Published:** 2022-11-06

**Authors:** Renata S. Fernandes, Raquel Gregório Arribada, Juliana O. Silva, Armando Silva-Cunha, Danyelle M. Townsend, Lucas A. M. Ferreira, André L. B. Barros

**Affiliations:** 1Faculty of Pharmacy, Universidade Federal de Minas Gerais, Av. Antônio Carlos, 6627, Belo Horizonte 31270-901, Brazil; 2Department of Drug Discovery and Pharmaceutical Sciences, Medical University of South Carolina, Charleston, SC 29425, USA

**Keywords:** doxorubicin, lipid–polymer hybrid nanoparticles, TPGS, antitumor activity, pH-sensitive, α-tocopherol succinate, drug delivery system

## Abstract

Doxorubicin (DOX) is an antineoplastic agent clinically employed for treating breast cancer patients. Despite its effectiveness, its inherent adverse toxic side effects often limit its clinical application. To overcome these drawbacks, lipid–polymer hybrid nanoparticles (LPNP) arise as promising nanoplatforms that combine the advantages of both liposomes and polymeric nanoparticles into a single delivery system. Alpha-tocopherol succinate (TS) is a derivative of vitamin E that shows potent anticancer mechanisms, and it is an interesting approach as adjuvant. In this study, we designed a pH-sensitive PLGA-polymer-core/TPGS-lipid-shell hybrid nanoparticle, loaded with DOX and TS (LPNP_TS-DOX). Nanoparticles were physicochemically and morphologically characterized. Cytotoxicity studies, migration assay, and cellular uptake were performed in 4T1, MCF-7, and MDA-MB-231 cell lines. Antitumor activity in vivo was evaluated in 4T1 breast tumor-bearing mice. In vitro studies showed a significant reduction in cell viability, cell migration, and an increase in cellular uptake for the 4T1 cell line compared to free DOX. In vivo antitumor activity showed that LPNP-TS-DOX was more effective in controlling tumor growth than other treatments. The high cellular internalization and the pH-triggered payload release of DOX lead to the increased accumulation of the drugs in the tumor area, along with the synergic combination with TS, culminating in greater antitumor efficacy. These data support LPNP-TS-DOX as a promising drug delivery system for breast cancer treatment.

## 1. Introduction

Breast cancer is the most common cancer in women. The new cases of patients diagnosed in 2022 is expected about 300,000 in the United States and 66,000 in Brazil [1,2]. Although great progress has been achieved in endocrine treatment and epidermal growth factor 2 (ERBB2)-targeted therapy, these treatments are not suitable for all types of breast cancer and chemotherapy still remains an essential treatment for preventing recurrence in many patients [3]. Doxorubicin (DOX) is an antineoplastic agent within the anthracycline and antitumor antibiotics family, clinically employed alone or in combination for treating various tumors. Despite being considered one of the most effective anticancer drugs, its inherent adverse toxic side effects on healthy tissues, particularly bone marrow and the cardiovascular system, are a critical problem that often limits clinical application [4,5]. In the attempt to overcome these drawbacks, nanoparticles have been extensively investigated as drug delivery systems to enhance antitumor efficacy and reduce adverse effects on normal tissues [4,6,7,8].

One of the approaches to overcome drawbacks are the development of lipid–polymer hybrid nanoparticles (LPNP) that unite favorable attributes of both liposomes and polymeric nanoparticles into a single delivery system. The hybrid nanoparticle is composed of a biodegradable hydrophobic polymeric core, a monolayer of phospholipids, and an outer corona layer made of PEG. The phospholipid shell contributes to its high biological compatibility, safety profile, and better loading capacity. The solid polymer core provides a structural framework that leads to enhanced mechanical stability, shape control, biodegradability, homogenous size distribution, and an extensive surface area. Moreover, PEG chains confer steric stabilization and reduce opsonization [9,10,11,12,13]. Poly(lactic-co-glycolic acid) (PLGA) is one of the more frequently used hydrophobic polymers for core formation in LPNP. PLGA is a copolymer composed of polyglycolic acid (PGA) and polylactic acid (PLA). The hydrolysis into monomers, which are readily metabolized via the Krebs cycle, render this polymer biodegradable, leading to minimal systemic toxicity associated. Both the United States FDA and European Medicine Agency (EMA) have already approved the use of PLGA in several drug delivery systems for humans [14,15].

Beyond the drug delivery system, combination therapy brings added value of improved therapeutic efficiency and diminished side effects. The use of adjuvants with antitumor properties and favorable safety profiles is an interesting choice in drug formulation. Alpha-tocopherol succinate (TS) is a derivative of vitamin E that has been shown to have both in vitro and in vivo anticancer effects by impacting proliferation, differentiation, and apoptosis [16]. D-α-Tocopheryl polyethylene glycol 1000 succinate (TPGS) is a water-soluble derivative of Vitamin E that is formed by esterification of TS with polyethylene glycol (PEG) 1000. TPGS acts as an emulsifier and enhances bioavailability of hydrophobic drugs because it has a large, bulky surface area. TPGS is approved by the FDA as a safe pharmaceutical adjuvant and has antitumor activity comparable to TS. Therefore, the use of TPGS as part of the lipid shell in LPNP is a promising alternative since it is possible to combine its pharmacological and pharmacotechnical properties [17,18,19].

In this work, we designed and generated a pH-sensitive polymer-core/lipid-shell hybrid nanoparticle, loaded with DOX and TS (LPNP-TS-DOX), for anticancer drug delivery and controlled release. Co-loading these two drugs, we take advantage of the ion-pairing formation between DOX and TS, which confers higher hydrophobicity to the drug that may improve affinity to polymeric core and pH-sensitivity to the system since the ion-pair is detached in an acidic pH. LPNP-TS-DOX was prepared by emulsion-solvent evaporation (ESE) method, and it was composed of PLGA as the polymer core and a mixture of lecithin and TPGS as a lipid shell. In vitro and in vivo properties of this novel nanosystem were evaluated. Cytotoxicity studies, migration assay, and cellular uptake were performed in three breast cancer cell lines. Antitumor activity in vivo was evaluated in 4T1 breast tumor-bearing model.

## 2. Materials and Methods

### 2.1. Material

Doxorubicin hydrochloride (DOX) was purchased from ACIC Chemicals (Brantford, ON, Canada). α-Tocopherol succinate (TS), α–Tocopherol polyethylene glycol 1000 succinate (TPGS), and Bovine Serum Albumin (BSA) were purchased from Sigma-Aldrich (São Paulo, Brazil). Poly(L-lactide-co-glycolide) 50:50 (PLGA) was obtained from Evonik Health Care (Essen, Germany). Soybean Lecithin was purchased from Lipoid GmbH (Ludwigshafen, Germany). Triethanolamine (TEA) was obtained from Merck (Darmstadt, Germany). RPMI1640 medium, Dulbecco’s Modified Eagle’s Medium (DMEM), Minimum Essential Medium (MEM), streptomycin, penicillin, and amphotericin B (PSA), fetal bovine serum (FBS), and trypsin EDTA (0.25%) were purchased from Gibco-Invitrogen (Grand Island, NE, USA). All other chemicals were analytical grade.

### 2.2. Preparation of LPAP

LPNP-TS-DOX was produced by the emulsion-solvent evaporation (ESE) method. Briefly, the organic phase was composed of PLGA 50:50, lecithin, TS, and TEA dissolved in dichloromethane, and the aqueous phase was formed by TPGS and DOX in purified water. The organic phase was introduced dropwise into the aqueous phase, under continuous agitation, for 5 min, with an Ultra Turrax T-25 homogenizer (Ika Labortechnik, Staufen, Germany), at 10,000 rpm. The samples were then sonicated at 20% amplitude for 10 min using a high-intensity ultrasonic processor (CPX 500 model, Cole-Palmer Instruments, Vernon Hills, IL, USA). The suspension was kept under magnetic agitation, at room temperature, for 5 h, for complete evaporation of dichloromethane. DOX was entrapped in the LPNP by the formation in situ of an ion-pairing between DOX and TS, as shown in Figure 1. LPNP-TS-DOX was purified using Amicon^®^ centrifugal devices (30 min at 3500 g).

### 2.3. Characterization

The LPNP-TS-DOX preparations were characterized by polydispersity index (PDI), mean hydrodynamic diameter, zeta potential, and nanoparticle tracking analysis (NTA). Additionally, LPNP-TS-DOX was evaluated for encapsulation efficiency (EE), drug content (%DL), and DOX in vitro release. Images of transmission electron microscopy (TEM) and transmission electron cryo-microscopy (Cryo-TEM) were used to evaluate morphological features [20].

### 2.4. Mean Diameter and Zeta Potential

Dynamic light scattering (DLS) at a fixed angle of 90° and 25 °C was used to measure the mean particle diameter using a Zetasizer Nano ZS90 (Malvern Instruments, Worcestershire, UK). DLS combined with electrophoretic mobility was used to measure the zeta potential at 25 °C. All measurements were performed in triplicate in 10-times diluted LPNP-TS-DOX in distilled water.

### 2.5. Drug Encapsulation Efficiency (%EE) and Drug Loading

The encapsulation efficiency and drug loading were determined by HPLC using the conditions published elsewhere [21]. A total of 4 mL of the suspension was ultra-filtrated, for 30 min at 3500 g, using centrifugal devices (Amicon^®^ Ultra–100 k, Millipore, Burlington, MA, USA). In these conditions, DOX passes through the membrane while LPNP remains in the filter. The non-encapsulated DOX was quantified in the filtrate while the LPNP-TS-DOX stuck in the filter was solubilized with 2 mL of DMSO for quantification. The encapsulation efficiency (EE) was calculated by using the following equations:%EE = [(CT − CF)/ CT] × 100 (1)
where: CT = total mass of doxorubicin in purified LPNP and aqueous phase (non-encapsulated), CF = mass of doxorubicin in aqueous phase (non-encapsulated).

The purified LPNP-TS-DOX suspension was completely dried in a 48 h-cycle of lyophilization (Modulyolyophilizer-Thermo Electron Corporation, Milford, MA, USA). The resultant content was weighted and resuspended in DMSO for DOX quantification. Drug loading (DL) was calculated by using the following equation:%DL = WD/ WNP × 100(2)
where WD = weight of doxorubicin in nanoparticles, WNP = weight of nanoparticles.

### 2.6. Nanoparticle Tracking Analysis (NTA)

NTA experiments were carried out using NanoSight NS300 & NTA 3.1 Analytical Software (Malvern Instruments, Worcestershire, UK). LPNP-TS-DOX were diluted (5000×) in ultrapure water and placed in the NanoSight sample chamber. The suspension was irradiated for 60 s at room temperature by a laser source and light scattering. A charge-coupled device camera was used to capture images.

### 2.7. Morphological Analysis

Morphological analysis of NLPs was performed on a transmission electron microscopy (TEM) and transmission electron cryo-microscopy (cryo-TEM) (Tecnai G2-12-FEI SpiritBiotwin 120 kV). For TEM images, LPNP-TS-DOX were rendered electron-dense by staining with 2% phosphotungstic acid solution and placed on a carbon-coated copper grid and dried under a heat lamp for 10 min prior to observation under TEM. For cryo-TEM images, samples were prepared using the plunge freezing technique whereby the samples were spread on a thin film across an EM grid and submerged in liquid ethane.

### 2.8. In Vitro Release of Dox

Dialysis was used to assess the release of DOX. Dialysis membranes with a size exclusion of 14 kDa and diameter of 21 mm (cellulose ester membrane; Sigma–Aldrich, St Louis, MO, USA) were used. The membranes containing 1 mL LPNP-TS-DOX were incubated with 100 mL of PBS (pH 7.4) and HEPES (pH 5.0) for 24 h at 37 °C. The control/free DOX was prepared in aqueous solution of DOX (1.0 mg/mL). Aliquots were removed at 1, 2, 4, 8, and 24 h for DOX quantification via HPLC. Values were plotted as the cumulative percentage of drug release.

### 2.9. LPNP-TS-DOX Stabality

#### 2.9.1. Colloidal Stability

The stability of the LPNP-TS-DOX was evaluated in different media. Two mL of LPNP-TS-DOX were ultrafiltrated in Amicon^®^ devices and resuspended in PBS buffer (pH 7.4) or culture media (MEM and RMPI supplemented with 10% of FBS). The solutions were incubated for 24 h, at 37 °C, under magnetic stirring at 75 rpm. At pre-determined time-points mean diameter, PDI, and zeta potential were measured.

#### 2.9.2. Colloidal Stability in Albumina

In order to evaluate the stability of LPNP-TS-DOX in presence of bovine serum albumin, a PBS solution of BSA was prepared (58.0 mg/mL). Then, 340 µL of the solution was added to 1 mL of LPNP-TS-DOX previously diluted (ratio NP:albumin 1:100), and incubated at 37 °C, under continuous stirring, at 75 rpm, for 1 h. The mixture was analyzed as mean diameter, PDI, and zeta potential, at time zero and after incubation.

#### 2.9.3. Storage Stability

The storage stability of the solution was evaluated. Fresh-made samples were put into capped glass containers, under N_2_ atmosphere, and stored at 2–8 °C, for 30 days, protected from the light. After 7, 15, and 30 days of storage, particle size, polydispersity index, and zeta potential were evaluated by DLS, and encapsulation efficacy was determined by HPLC.

### 2.10. In Vitro Studies

#### 2.10.1. Cell Culture

The human breast cancer cell lines, MDA-MB-231 (ATCC HTB-26) and MCF-7 (ATCC CRL-3435) were cultured in DMEM and MEM, respectively. The murine breast cancer cell line 4T1 (ATCC CRL-2539) was cultured in RPMI. For all cells, the media was supplemented with 10% FBS and 1% PSA. Cells were maintained at 37 °C and 5% CO_2_ in a humidified atmosphere.

#### 2.10.2. Cytotoxicity Studies

Human (MCF-7, MDA-MB-231) and murine (4T1) breast cancer cells were seeded in 96-well plates at a density of 1 × 10^4^ cells/well and 5 × 10^3^ cells/well, respectively 24 h prior to treatment. Cells were exposed to DOX, LPNP-TS-DOX, and LPNP-blank (0.01–10 µM), for 48 h. Cell viability was assessed using sulforhodamine B (SRB) assay. Cells were fixed with 10% trichloroacetic acid (TCA) and then stained with SRB for 30 min. Unbound SRB was withdrawn using 1% acetic acid while the protein-bound SRB was dissolved in 10 mM of Tris-Base [tris(hydroxymethyl) aminomethane] solution. The optical densities (OD) were measured at 510 nm on a microplate spectrophotometer Spectra Max Plus 384 (Molecular Devices, Sunnyvale, CA, USA).

#### 2.10.3. Migration Assay

To study the bi-dimensional migration, MDA-MB-231, MCF-7, and 4T1 cells were seeded in 12-well plates (3.0 × 10^5^, 1.5 × 10^5^, 2.0 × 10^5^ cells/well, respectively) and incubated at 37 °C for 24 h. A 200 uL pipette tip was used to generate a straight wound in each well and immediately imaged as the reference of “0 h,” and the “zero wound” using a microscope AxioVert 25 with a connected Axio Cam MRC camera (Zeiss, Oberkochen, Germany). The cells were then treated with 1 mL fresh media (control) or media containing DOX (free or encapsulated into LPNP). All plates were, then, incubated for 24 h at 37 °C. Throughout the assays, the cells were serum starved in 1% FBS to accurately differentiate that cell migration, but not proliferation, was responsible for the wound closures. Afterward, cells were then fixed using formaldehyde 4% for 10 min and images were obtained in phase contrast and assessed using the MRI Wound Healing Tool plugin for the free version of Image J 1.45 software (National Institutes of Health, Bethesda, MD, USA). The % wound-healing was calculated as follows:% wound healing = 100 − (area of treated wound × 100)/ area of the “zero wound”(3)

#### 2.10.4. Cellular Uptake

Cellular uptake is a powerful tool to measure the ability of the system or drug to enter the cell where it can accumulate. To evaluate cellular uptake of DOX, 1.0 × 10^6^ of MDA-MB-231, 4T1, and MCF-7 cell lines were seeded in a 12-well plate and incubated at 37 °C for 24 h before treatment. The cells were then exposed to treatment with DOX or LPNP-TS-DOX, at a concentration of 3 µM. After 1, 2, 4, and 8 h, the cells were harvested by trypsin and centrifuged (5600× *g*, 5 min). The supernatant was removed, and the cells were washed with PBS, to remove adsorbed doxorubicin. Afterward, proteins were precipitated using isopropyl alcohol in an ultrasonic bath for 5 min. The suspension was centrifuged (5600× *g*, 5 min) and the supernatant was separated by HPLC for DOX quantification as previously described.

### 2.11. In Vivo Studies

#### 2.11.1. Animals and Tumor Cell Inoculation

The central vivarium of Federal University of Minas Gerais-Brazil provided 8-week-old female BALB/c mice (18–22 g). The in vivo experiments were approved by the local Ethics Committee for Animal Experiments (CEUA/UFMG), under the protocol #179/2019 (Approval date 16 September 2019). Aliquots (100 µL) of 1.0 × 10^6^ 4T1 cells were administrated by subcutaneous injection into the right flank of each animal. Tumor cell growth was monitored over 7 days and did not exceed a volume of 100 mm^3^. Animals were housed in cages in a controlled environment (25 °C, 30–70% humidity). Mice were maintained in a 12 h light-dark cycle and provided free access to food and water.

#### 2.11.2. Liposome Preparation

Doxorubicin-loaded liposomes (similar to the commercially available) were used as a comparative group in the in vivo antitumor assay. Liposomes were prepared using the lipid film hydration method (Bangham, 1965), followed by size calibration. Briefly, chloroform aliquots of hydrogenated soybean phosphatidylcholine (HSPC), cholesterol, and distearoyl-phosphoethanolamine-polyethilenglycol2000 (DSPE-PEG2000) (5.8: 3.7: 0.5 molar ratio, at 20 mM lipid concentration) were added to a round-bottom flask and the chloroform was evaporated at low pressure. Then, 300 mM ammonium sulfate solution (pH 7.4) was used to hydrate the liposome film at room temperature. The particle size was calibrated by extrusion using polycarbonate membranes of 0.4 μm, 0.2 μm and 0.1 μm, 5 cycles per membrane, using the Lipex Biomembranes extruder, Model T001 (Vancouver, BC, Canada). Ultracentrifugation at 150,000× *g*, 4 °C for 120 min was used to remove ammonium sulfate (Optima^®^ L-80XP, Beckman Coulter, Indianapolis, IN, USA). HEPES-saline buffer (pH 7.4) was used to resuspend the pellet of liposomes. Remote encapsulation was performed with a DOX solution (2 mg/mL) for 2 h at 4 °C. Nonencapsulated DOX was removed by ultracentrifugation. The liposomes were characterized and reached size (131 ± 4 nm), PDI (0.02 ± 0.01), zeta potential (−1.7 ± 0.4 mV), and drug encapsulation efficiency (90.1 ± 2%) suitable for in vivo studies and consistent with previous studies [22].

#### 2.11.3. Antitumor Activity

Seven days after inoculation (described above) mice were randomly assigned into four groups (n = 6). Group 1: control/LPNP-blank; group 2: free DOX; group 3: LPNP-TS-DOX; and group 4: Liposome-DOX. For all groups, 5 mg DOX/kg/day was injected by tail vein every other day for a total of 4 doses totaling 20 mg/kg. Tumor volume was measured every other day with calipers and calculated as follows:V = (d1)^2^ × d2 × 0.5(4)
where d1 and d2, represent the smaller and larger diameter, respectively.

The relative tumor volume (RTV) and inhibition ratio (IR) were calculated on the last day, as follows:RTV = Tumor volume D8/ Tumor volume D0(5)
where D8 = Last day of the experiment and D0 = Day of the beginning of the treatment.
IR = 1 − (Mean RTV of drug-treated group/Mean RTV of control group) × 100

### 2.12. Statistical Analysis

Statistical analyses were performed using GraphPad PRISM, version 5.00 software (GraphPad Software Inc., La Jolla, CA, USA). Differences between two groups were evaluated using a Student’s *t* test. Differences among experimental groups were assessed using the one-way analysis of variance (ANOVA), then Tukey’s test was run. Statistical significance was considered when *p* values were <0.05.

## 3. Results

### 3.1. Characterization

LPNP-TS-DOX was evaluated by average size, PDI, zeta potential, %EE, and %DL as demonstrated in Table 1. The mean diameter of about 150 nm and a PDI of 0.25 indicates that the hybrid nanoparticle was formed successfully and homogeneously. The highly negative zeta potential achieved in ultrapure water is likely due to the presence of TS excess that is deprotonated during the nanoparticle formation. The encapsulation efficiency was around 98%, while drug loading was about 6% and drug:polymer ratio used was 1:4.6.

LPNP-TS-DOX particle concentration and size distribution (i.e., the mean diameter in which 10, 50, and 90% of the particles are smaller) were assessed by NTA and the values are shown in Table 2. It can be observed that the mean size of the particles is about 115 nm, and 90% of them are under 159 nm, which is consistent with DLS analysis.

Representative TEM and cryo-TEM images of LPNP-TS-DOX are shown in Figure 2. We used two different types of sample preparation—negative staining (Figure 2A) and plunge freezing (Figure 2B) for morphological evaluation. In both techniques, the particles showed round, smooth surfaced, and nanosized particles. It is possible to observe a thin layer surrounding the inner bright portion, indicating the presence of a polymeric core, surrounded by the lipid shell. During the evaporation of the organic solvent, the polymeric core is formed, and the lipids assemble around the polymer forming the shell at the same time [13,23].

In vitro drug release data are shown in Figure 3. DOX release from LPNP-TS was evaluated in two pH values. At pH 7.4, the DOX release from the nanoparticles (blue) reached 40% in 24 h, while in an aqueous solution DOX (red) releases more than 90% in only 4 h. At pH 5.0, about 64% of DOX was released from LPNP-TS-DOX (green) in 2 h, reaching a maximum of 73% in 24 h. These data indicate the pH-sensitivity of the formulation.

### 3.2. LPNP-TS-DOX Stability

#### 3.2.1. Colloidal Stability

LPNP-TS-DOX was tested in different media, such as PBS, RPMI, and MEM to evaluate colloidal stability. The stability parameters are summarized in Figure 4. We observed that the LPNP-TS-DOX showed excellent stability, in all three evaluated media, without any significant change in the particle mean diameter, PDI, and zeta potential in 24 h.

#### 3.2.2. Stability in Albumin

The stability of LPNP-TS-DOX in the presence of BSA was evaluated by DLS, Figure 5. The nanoparticles and BSA were incubated together, at 37 °C, under shaking for one hour. In the mixture (LPNP-TS-DOX + BSA) curve (green line), it is possible to observe the presence of a well-defined peak at about 20 nm, corresponding to BSA. Moreover, the peak of the nanoparticle is the same as its characteristic single peak measured alone (red line). DLS demonstrated that LPNP-TS-DOX had a weak interaction with BSA and were expected to be stable in serum.

#### 3.2.3. Storage Stability

LPNP-TS-DOX were stored in a refrigerator (2–8 °C), protected from light, in order to assess the short-term stability of the formulation. The mean diameter, PDI, zeta potential, and %EE were evaluated at different time points within 30 days. Table 3 represents the mean diameter, PDI, zeta potential, and %EE values over time. It can be noted that the mean size did not change over time. LPNP maintained doxorubicin encapsulated into the polymer core, maintaining %EE over 98% in all evaluated time-points. Furthermore, there was no significant change in PDI and zeta potential at all evaluated time. Furthermore, no aggregation, precipitation, or phase separation were observed during the evaluation.

### 3.3. In Vitro Studies

#### 3.3.1. Cytotoxicity Studies

Cytotoxicity assays were carried out against human (MDA-MB-231 and MCF-7), and murine (4T1) breast cancer cell lines. The cells were incubated with LPNP-TS-DOX, free DOX, and LPNP-blank at different concentrations and cell viability was assessed (Figure 6). First, it is important to highlight that the LPNP-blank did not induce cytotoxic effects, with ~ 100% viability at all evaluated concentrations. The LPNP-TS-DOX formulation showed significantly higher cytotoxicity (*p* < 0.05) at higher concentrations for 4T1 and MCF-7 cell lines. However, there was no difference between treatment groups in the MDA-MD-231 cells. These data indicate that the incorporation of drug into the nanoparticles does not impair its activity but, in contrast, leads to higher cytotoxicity in two of the cell lines tested.

#### 3.3.2. Migration Assay

Cell migration might be measured in a wound-healing assay by observing the wound closure in a monolayer cell culture. In this study, wound closures were evaluated 24 h after exposure to 500 nM of DOX as LPNP-TS-DOX or free DOX. A control group with no treatment was also evaluated but, as any drug was applied, the wounds completely close during the study and wound-area calculation was not possible for this group. Representative phase-contrast photomicrographs of the scratches after 24 h exposure to the treatments are shown in Figure 7.

As summarized in Table 4, there were no differences between the treatments on the percentage of cell migration for the MDA-MB-231 cell line. These data are consistent with the viability assays shown prior. However, in both MCF-7 and 4T1 cell lines, the encapsulation of DOX into the LPNP, along with the combination with TS, probably increased the toxicity of the treatment, significantly impairing the cell migration compared with the free DOX group (*p* < 0.05).

#### 3.3.3. Cellular Uptake

Cellular uptake studies in 4T1, MDA-MB-231, and MCF-7 cell lines was evaluated for LPNP-TS-DOX and free DOX (Figure 8). The cells were incubated with 3 µM of each formulation, for 8 h. HPLC analysis showed that DOX uptake was time-dependent. The cellular uptake studies are consistent with cytotoxicity/viability. The LPNP-TS-DOX formulation did not show a gain in cellular uptake in MDA-MB-231 and MCF-7 cell lines relative to free drug. However, for 4T1 cells, the cellular uptake for the nanosystem was significantly higher than free DOX, for all evaluated time points, which is consistent with the cell viability assay.

#### 3.3.4. Antitumor Activity

The antitumor activity of LPNP-TS-DOX and free DOX was tested using mice with an implanted breast tumor (4T1). LPNP-TS-DOX and Liposome-DOX were freshly prepared and characterized before the treatments. Results showed significant differences in the tumor volume among the control group versus DOX-treated groups (*p* < 0.05), as demonstrated in Figure 9A. Noteworthy, the antitumor activity of LPNP-TS-DOX was more effective than in the other treatment groups (*p* < 0.01). Specifically, the LPNP-TS-DOX group was more capable of inhibiting tumor growth than the other DOX treatment groups (*p* < 0.05). These findings indicate that LPNP-TS-DOX therapy was more successful than free DOX. Tumor volume data were confirmed by the RTV values (Figure 9B). Moreover, the IR for free DOX, Liposome-DOX, and LPNP-TS-DOX were 47%, 64%, and 86%, respectively, corroborating the effectiveness of the treatment with the hybrid nanoparticles. The body weight was also monitored every other day since the beginning of the treatment. As can be seen in Figure 9C, body weight variation was statistically the same for all dox-treated groups before euthanasia.

## 4. Discussion

The search to overcome the limitations of conventional chemotherapy provides alternative approaches to drug delivery systems, among them the LPNP that combines the advantages of liposomes and polymeric nanoparticles into a single robust platform [24]. Due to the polymeric core and the lipid shell LPNP have high structural reliability, storage stability, controlled release profile, and high biocompatibility [25]. In this work, we successfully designed and generated a pH-sensitive hybrid nanoparticle loaded with doxorubicin and TS. By using a simple method, the inner polymer core is formed by PLGA, which entraps DOX and TS. The lipid-surrounded layer, formed by lecithin and TPGS, confers biocompatibility and supports drug retention inside the polymer core. TPGS also provides a PEG chain that prolongs in vivo circulation time and helps in steric stabilization [26].

The physical properties of the delivery system were evaluated. LPNP-TS-DOX presented a mean diameter of about 150 nm (measured by DLS and NTA), PDI below 0.3 (0.25), which demonstrates a homogeneous distribution of the particles. From TEM and cryo-TEM images, it is possible to observe a thin layer surrounding the inner bright portion, indicating the formation of the core/shell system. These characteristics favor LPNP-TS-DOX accumulation in the tumor area by transcytosis and EPR effect. Transcytosis is a metabolically active process that requires endothelial cells to rearrange their cytoskeleton and cell membrane. Thus, nanoparticles can reach the tumor area by transport through the cytoplasm or uptake by vesicles that are formed in the process. On the other hand, EPR effect is a well-known phenomenon that allows the passive accumulation of nanoscale particles (from 10 to 500 nm) inside the tumor area, due to gaps formed in their blood vessel wall. These two mechanisms—passive transport through gaps or active transport through trans-endothelial pathways—account for the nanoparticle accumulation in the tumor area [27,28,29,30].

It was possible to achieve a high value of encapsulation efficiency (98%), which is surprisingly high for DOX-loaded polymeric core-based nanoparticles. Several DOX-loaded LPNP studies report an encapsulation efficiency of about 50%, even when the lipophilic DOX base was used [5,6,31,32]. The outstanding result achieved in this work is explained due to the ion pair formation between TS and DOX that increases the hydrophobicity of DOX, keeping the drug entrapped into the matrix [21,33,34]. The ion-pair formation could also explain the slow release of DOX from LPNP-TS-DOX at physiological pH and faster release in acidic pH. Once at an acidic pH, the carboxyl acid of TS could be protonated, disrupting DOX-TS interaction and consequently DOX release. The development of a stimuli-responsive drug delivery system is interesting since the tumor microenvironment exhibits different features compared to normal tissues, such as low pH due to elevated levels of lactic acid caused by poor oxygen perfusion. Moreover, the lower release of the drug in physiological pH could minimize its side effects in healthy tissues [31].

The developed nanosystem proved to be highly stable in several conditions. Storage stability showed that the nanoparticles remain stable for up to 30 days, keeping DOX entrapped into the polymer core, maintaining %EE at about 98%. LPNP-TS-DOX had a weak interaction with BSA, showing no size alteration. Moreover, it exhibited high stability in different media, such as PBS, RPMI, and MEM, for up to 24 h. The highly negative zeta potential observed can contribute to LPNP stability, due to electrostatic repulsion. Moreover, the use of TPGS might contribute to improving the colloidal stability, since incorporating suitable amounts of additional surfactants along with the phospholipids, confers stability to the system. Reasonably, the projection of the long and bulky PEG chain of the TPGS enhances stability as compared to the small choline head group of lecithin [5,13,23,35]. High serum stability is important to acquire prolonged circulation time and improved accumulation at the tumor site via the EPR effect or via transcytosis.

A panel of three breast cancer cell lines was used to evaluate the LPNP-TS-DOX behavior in cytotoxicity, migration, and cellular uptake. Cytotoxicity studies showed a significant reduction in cell viability for the 4T1 cell line, especially at higher concentrations, which is interesting for antitumor drugs since, for cancer therapies, is necessary to kill the greatest number of cells [36]. The gain in cytotoxicity might be explained by the cellular uptake results. It could be noted cellular uptake for LPNP-TS-DOX was significantly higher than free DOX in this cell line. Greater cell internalization leads to a greater supply of drug, which might increase cell cytotoxicity [37].

Although several processes might be involved in the metastasis event, one of the critical features that cells should evolve are their capacity for migration and tissue invasion from the primary tumor. Therefore, the migration assay herein showed that LPNP-TS-DOX was able to significantly impair cell migration in both the 4T1 and MCF-7 cell lines. These results emphasize the benefits of the encapsulation into the nanosystem since the presence of metastasis, besides being a limiting factor for the treatment, it also indicates poor prognosis and directly affects patient survival [38,39].

The 4T1 breast tumor-bearing BALB/c mouse model is a very usual experimental animal model to mimic late-stage (IV) human breast cancer [40]. DOX treatments were highly effective against 4T1 tumor cells, nevertheless, the encapsulation into LPNP was able to enhance even more its antitumor activity. The accumulation on the tumor site along with higher cell internalization, and the pH-triggered payload release leads to the increased accumulation of DOX in the tumor area, culminating in greater antitumor efficacy [31] (MEN et al., 2019). Additionally, the combination between DOX and TS (free and in TPGS form) may contribute to increasing antitumor efficacy, since its antitumor activity is already reported. The use of two or more drugs, at the same treatment schedule, has demonstrated promising outcomes in terms of improving therapeutic efficiency and reducing toxic effects [16,34,41,42].

The results obtained to-date are in agreement with those previously reported in the literature using nanoparticles as drug delivery systems. Several groups have developed DOX encapsulated hybrid-nanoparticles [43,44,45]. Hao and co-workers developed TPGS 2K micelles loaded with DOX. Li and co-workers used a PLA-TPGS copolymers to encapsulate DOX. Yang and co-workers developed a star-shaped polymer based on DOX, TPGS, and β-cyclodextrin. All three works evaluated the nanosystem against MCF-7 breast tumor cell lines. Results showed that the DOX-loaded TPGS 2K micelles and star-shaped nanoparticles displayed significantly higher antitumor activity compared to free DOX. In addition, the doxorubicin encapsulation into the PLA-TPGS based- nanoparticles leads to a higher accumulation of the drug into the nucleus which results in a significant increase in cytotoxicity [43,44,45]. In each of these studies, the polymer had to be synthesized before becoming a component of the nanoparticle. Our design and generation strategy offers advantages. An important one is the absence of synthesis steps in the preparation of LPNP-TS-DOX, which facilitate the preparation and make the scaling up process faster and easier.

## 5. Conclusions

In this study, we developed and characterized pH-sensitive PLGA-TPGS-based hybrid nanoparticles loaded with doxorubicin and TS. This novel nanoplatform presented higher cytotoxicity, tumor uptake, and significantly reduced cell migration in vitro. In vivo antitumor activity showed that the nanosystem was more effective in controlling tumor growth than other DOX treatments. More preclinical studies still have to be performed, however, the data obtained so far already indicate that the nanosystem is a promising platform for the treatment of breast cancer.

## Figures and Tables

**Figure 1 pharmaceutics-14-02394-f001:**
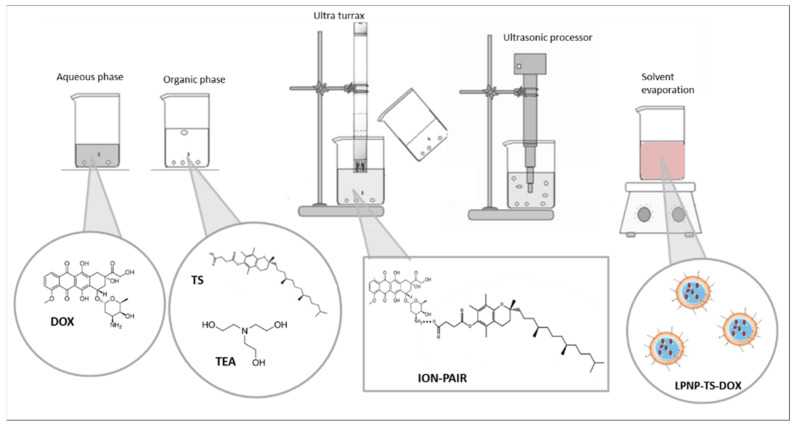
Schematic representation of LPNP-TS-DOX preparation method and formation in situ of the ion pair between DOX and TS.

**Figure 2 pharmaceutics-14-02394-f002:**
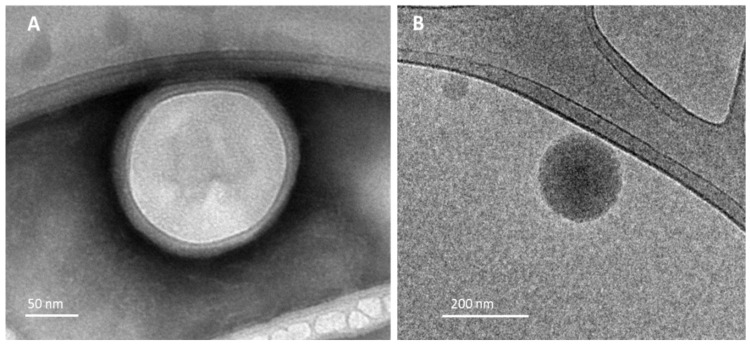
TEM images of LPNP-TS-DOX by (**A**) negative stain method (**B**) plunge freezing method.

**Figure 3 pharmaceutics-14-02394-f003:**
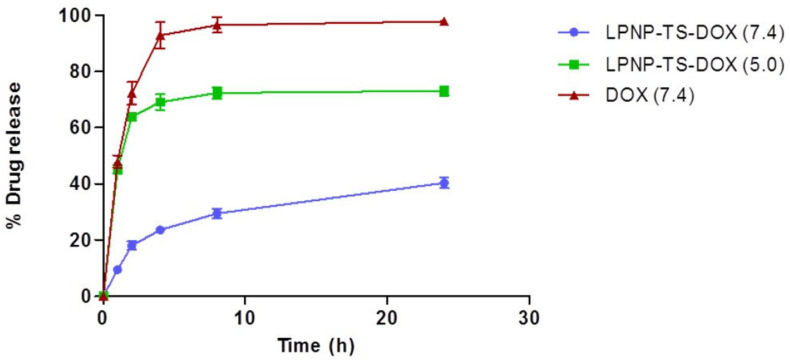
Drug release achieved for free DOX, in PBS buffer (pH 7.4), and encapsulated DOX, in PBS (pH 7.4) and HEPES (pH 5.0) buffers, at 37 °C, by dialysis method. Results are shown by the mean ± standard deviation (n = 3).

**Figure 4 pharmaceutics-14-02394-f004:**
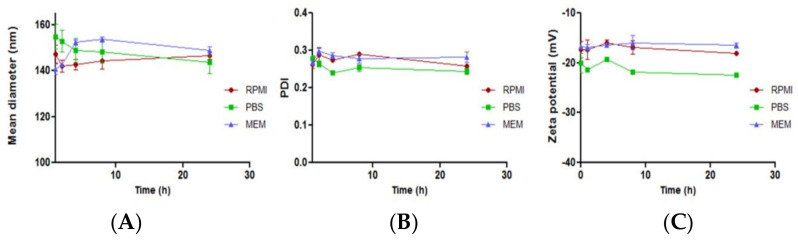
Effect on the (**A**) particle size, (**B**) PDI, and (**C**) zeta-potential of LPNP-TS-DOX after incubation in PBS (green line), RPMI (red line), and MEM (blue line), up to 24 h, at 37 °C. Results are shown by the mean ± standard deviation (n = 3).

**Figure 5 pharmaceutics-14-02394-f005:**
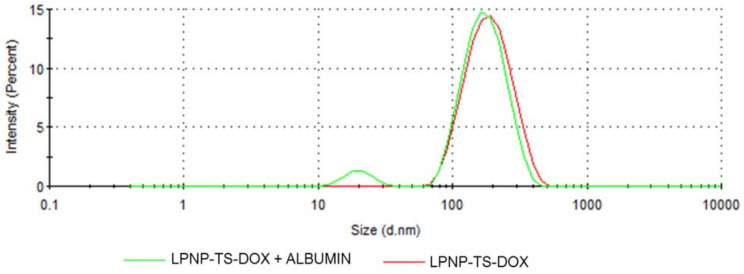
Stability of LPNP-TS-DOX in presence of PBS with BSA 5% w/v. Size distribution of NLP-TS-DOX (green line) and LPNP-TS-DOX + BSA (red line) were measured by dynamic light scattering.

**Figure 6 pharmaceutics-14-02394-f006:**
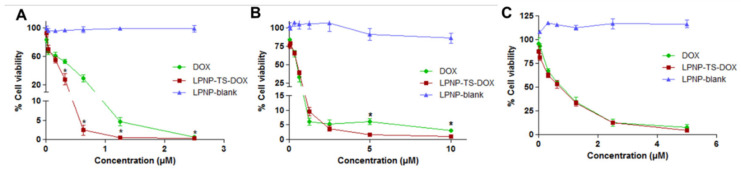
Cytotoxicity of NLP-blank, NLP-TS-DOX, and free DOX against (**A**) 4T1, (**B**) MCF-7, and (**C**) MDA-MB-231 cells, after incubation for 48 h. Data are expressed by the mean ± standard deviation of the mean (n = 3). Asterisks indicate statistically significant differences between the complexes at the same time point (*p* < 0.05).

**Figure 7 pharmaceutics-14-02394-f007:**
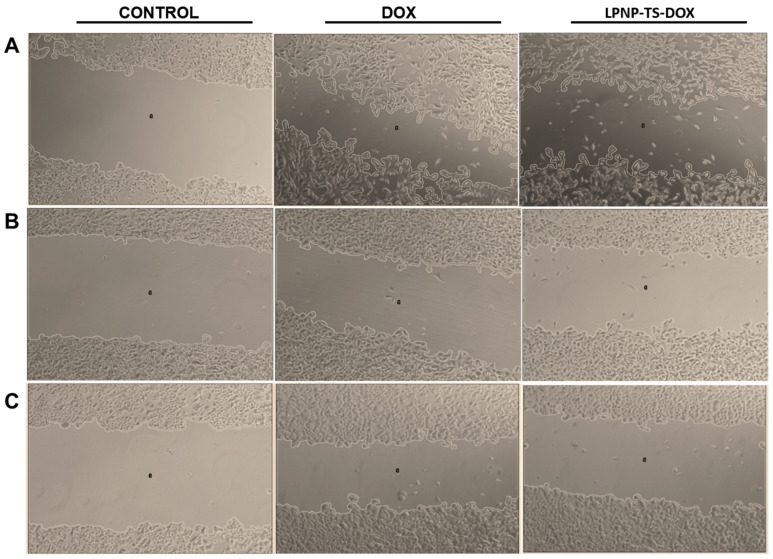
Phase-contrast photomicrographs of the “zero wound” (control) and wounds of (**A**) 4T1, (**B**) MDA-MB-231, (**C**) MCF-7 cell lines, exposed to 500 nM of DOX(free or encapsulated into LPNP), for 24 h at 37 °C. Amplification 5×.

**Figure 8 pharmaceutics-14-02394-f008:**
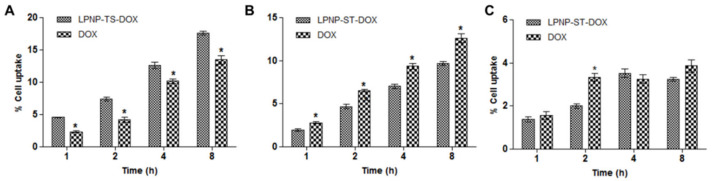
Cellular uptake of free DOX and NLP-TS-DOX evaluated in (**A**) 4T1, (**B**) MDA-MB-231, and (**C**) MCF-7 cell lines. Results are presented as the mean of at least three independent experiments and error bars represent SD. Statistically significant differences between the complexes at the same time point are indicated by asterisks (*p* < 0.05).

**Figure 9 pharmaceutics-14-02394-f009:**
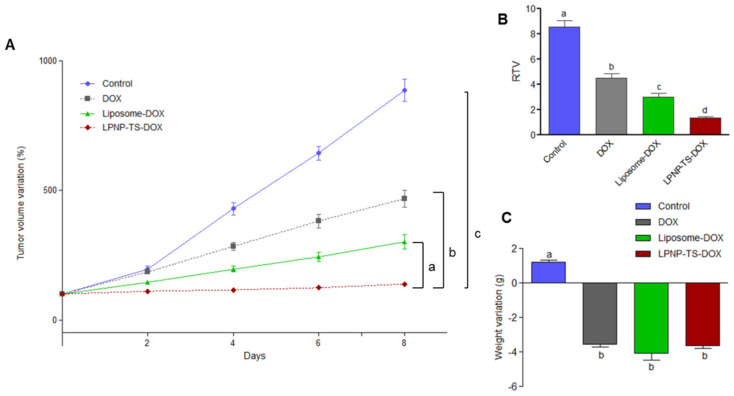
(**A**) Antitumor effect of blank-NLC (control), free DOX, Liposome-DOX, and NLP-TS-DOX on the growth of 4T1 tumor implanted into BALB/c mice. Each animal received by IV injection a dose of 5 mg/kg per day every other day (total of four administrations). Results are presented by the mean ± standard deviation (n = 7). Growth curves were analyzed by One-way ANOVA, followed by Tukey’s test. a Represents significant differences (*p* < 0.01) between NLP-TS-DOX and Liposome-DOX treatments. b Represents significant differences (*p* < 0.001) between NLP-TS-DOX and DOX treatments. c Represents significant differences (*p* < 0.001) between NLP-TS-DOX and Control treatments. (**B**) RTV analysis of Control, DOX, Liposome-DOX, and NLP-TS-DOX groups. Results are presented by the mean ± standard deviation (n = 7). Different letters indicate significant differences among groups (*p* < 0.05). (**C**) Body weight variation between D0 and D8. Results are presented by the mean ± standard deviation (n = 7). Different letters indicate significant differences among groups (*p* < 0.05).

**Table 1 pharmaceutics-14-02394-t001:** Mean diameter, polydispersity index (PDI), zeta potential, encapsulation efficiency (%EE), and drug loading (%DL) for LPNP-TS-DOX.

Mean Size (nm)	PDI	Zeta Potential	%EE	%DL
152.97 ± 2.77	0.251 ± 0.013	−29.6 ± 1.41	98.10 ± 0.89	6.03 ± 0.17

Data are expressed by the mean ± standard deviation of the mean (n = 3).

**Table 2 pharmaceutics-14-02394-t002:** NTA results: mean size, distribution (D10, D50, D90), and particle concentration of LPNP-TS-DOX.

Parameters (nm)				Particles/mL
Mean Size	D10	D50	D90	
114.4 ± 1.6	78.0 ± 1.3	102.0 ± 1.8	159.0 ± 4.3	1.07 × 10^12^

**Table 3 pharmaceutics-14-02394-t003:** Mean size, PDI, zeta potential, and %EE of LPNP-TS-DOX stored at 2–8 °C, protected from the light, for up to 30 days.

Day	Mean Diameter (nm)	PDI	Zeta Potential (mV)	%EE
0	152.97 ± 9.21	0.26 ± 0.02	−30.13 ± 3.00	98.65 ± 0.20
7	167.85 ± 10.10	0.25 ± 0.02	−29.64 ± 1.41	99.53 ± 0.12
15	146.35 ± 8.31	0.24 ± 0.01	−29.5 ± 1.85	99.43 ± 0.08
30	152.4 ± 11.78	0.23 ± 0.02	−25.22 ± 1.30	99.37 ± 0.10

Results are shown by the mean ± standard deviation (n = 3).

**Table 4 pharmaceutics-14-02394-t004:** Percentage of cell migration in relation to the control for 4T1, MDA-MB-231 and MCF-7 cell lines, when exposed to 500 nM of free or encapsulated DOX, for 24 h. The results were expressed as mean ± standard deviation (n = 3). Asterisks indicate statistically significant differences between the treatment groups in the same cell line (*p* < 0.05).

Treatment	4T1	MDA-MB-231	MCF-7
DOX	45.61 ± 8.76	27.15 ± 7.16	28.13 ± 7.01
LPNP-TS-DOX	34.54 ± 1.80 *	28.07 ± 13.87	17.15 ± 7.04 *

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
