# Peer review of "In Vitro and In Vivo Effect of pH-Sensitive PLGA-TPGS-Based Hybrid Nanoparticles Loaded with Doxorubicin for Breast Cancer Therapy"

_pharmaceutics, 2022, doi:10.3390/pharmaceutics14112394_

Round 1
Reviewer 1 Report
1. In the introduction, please highlight the novelty of this work and advantages of TPGS-PLGA-NP formulated in this work.
2. According to the preparation of LPNP-TS-DOX, DOX was mixed in the aqueous phase. How was the drug entrapped in the NP? By which mechanism?
3. In section 2.10.3, Line 199, please specify what drug is.
4. Why did the authors use two different methods to determine the size of NP? Please elaborate the differences between these two methods and which one is more reliable?
5. Figure 1, please provide image with more than one particle.
6. In section 3.2.2 the authors mentioned that a peak at 20 nm was referred to albumin. Please measure BSA alone to confirm this explanation.
7. The stability of NP at other temperatures such as room temperature and at 40 C should be performed to accelerate the long-term instability of the NPs.
8. Line 360, please specify the concentrations of NP used in this assay and discuss if the anti-migratory effect is due to the toxicity of the NP or not.
9. Table 4, please define a and b.
10. Line 391, please define "IR".
11. For antitumor activity, did the authors normalize concentration of encapsulated DOX for each formulation?
Author Response
Please find the response letter attached.

Reviewer 2 Report
In the manuscript
"In Vitro and In Vivo Effect of pH-Sensitive PLGA-TPGS-Based 2 Hybrid Nanoparticles Loaded with Doxorubicin for Breast Can-3 cer Therapy"
an approach for using hybrid nanoparticles fro improved delivery of the anti-cancer drug doxorubicine is presented. The biological data demonstrate, that the applied nanoparticles show activity, but, unfortunately, there is only very little information given on the nanoparticle technology, and the rationale and justification of the selected particles. The value of the manuscript would greatly improve, if such thourough information could be included. The information on the foundation of the proposed improved particle architecture should be given as otherwise the value to the community would be very limited. As well, results from control expeiments should e given (e.g. PLGA without shell) to allow insight into the relevance of the approach. If the authors could add such data and information the general interest for the manuscript would greatly improve.
Author Response

(The authors gave the same response as above.)

Reviewer 3 Report
This paper focuses on the synthesis and characterisation of polymer-lipid hybrid nanoparticles, based on PLGA and tocopherol succinate, as doxorubicin (DOXO) delivery system. This formulation may show interesting physicochemical ad biological properties for DOXO delivery in tumors, although many similar nanoparticle systems have been proposed and presented in literature in the last years. The manuscript is clear and well written, however hypothesis and conclusions are not always supported by a correct data presentation and interpretation. I therefore propose major revision before accepting this work for publication.
Detailed comments:
- Title & Abstract. The nomenclature used in the abstract should match the one used in the title. For instance, PLGA and TPGS are not introduced. The title is also misleading, as the polymer/formulation is not pH dependent on its own, but DOX typically has a pH dependent release simply because of its solubility (pKa).
- Materials and methods. The mechanism behind DOX encapsulation in the nanoparticles is unclear, since DOX is water soluble while the PLGA is hydrophobic. Generally, DOX can be loaded into PLGA when it is deprotonated, however this doesn’t seem to be the case. The equation used to calculate the encapsulation efficiency (pag 3) is wrong (check brackets): the correct one should be %EE=(CT-CF)/CT*100, where CT and CF are expressed in mass (not as concentration). The %EE values shown in table 1 and 3 are too high: 98% means that almost all DOX has been loaded, which is unlikely, and mostly impossible to obtain with polymer nanoparticles, unless DOX crystallized, or it is chemically conjugated. Drug loading (DL, equation 2) should be calculated as mg/mg and multiplied by 100 if expressed in %.
- Please correct typos in the text (see for example ‘Stabality’ and ‘Albumina’).
- Table 1: the low values of z-potential should be commented, since these particles are supposed to be PEGylated, and PEG typically shifts Z-potential to nearly zero, depending on its grafting density. Table 2: please explain what D10, D50, D90 mean.
- Figure 1. TEM showed only a single nanoparticle, however a population of nanoparticles should be presented too. A full DLS size distribution curve should also be provided.
- Figure 2. DOX release curve at pH=5 shows a plateau at 64%. Please explain why the release stopped at that value without showing a sustained release instead.
- Paragraph 3.2.1. please explain what PBS, RPMI and MEM are.
- Figure 4. Albumin has a hydrodynamic diameter of 6-7 nm. Please explain why the size distribution curve show a small peak at 20 nm instead.
- Figure 5. Caption should also provide the incubation time to for the cytotoxicity test.
- Figure 6. In this test, ‘zero-wound’ images are shown, however a negative control - with the wound treated with (only) medium but for the same incubation time - is missing. Without this negative control it is impossible to draw any conclusion on this test. Moreover, according to the images (and Table 4) I don’t see a ‘significant’ difference between the wound treated with nanoparticles and the one treated with free DOX.
- Paragraph 3.3.4. Please explain what IR and RTV mean.
Author Response

(The authors gave the same response as above.)

Round 2
Reviewer 1 Report
Please include all changes into the revised manuscript.
Reviewer 3 Report
I'm fine with the revision made, the paper in ready for publication.